

# Surface CO₂ Gradients Challenge Conventional CO₂ Emission Quantification in Lentic Water Bodies under Calm Conditions.

Patrick Aurich[1], Uwe Spank[2], Matthias Koschorreck[1]

[1]Department Lake Research, Helmholtz-Centre for Environmental Research – UFZ, Brückstraße 3a, 39114 Magdeburg, Germany

[2]Chair of Meteorology, Institute of Hydrology and Meteorology, Faculty of Environmental Sciences, Technische Universität Dresden, Pienner Straße 23, 01737 Tharandt, Germany

*Correspondence to*: Patrick Aurich (patrick.aurich@ufz.de)





**Abstract.** Lakes are hotspots of inland carbon cycling and are important sources of greenhouse gases (GHGs), such as carbon dioxide ($CO_2$). The significant role of $CO_2$ in global carbon cycle makes quantifying its emission from various ecosystems, including lakes and reservoirs, important for developing strategies to mitigate climate change. The thin boundary layer method is a common approach to calculate $CO_2$ fluxes from $CO_2$ measurements in both the water and the air, and wind speed. However, one assumption for the TBL method is a homogeneous $CO_2$ concentration between the measurement depth and the water surface, where gas exchange takes place. This assumption might not be true under calm conditions, when microstratification below the surface slows vertical exchange of gases. We used a floating outdoor laboratory to monitor $CO_2$ concentrations in 5 cm and 25 cm depth, $CO_2$ concentration in the air, wind speed, and water temperature profiles for one week in Bautzen Reservoir, Germany. While we found homogeneous $CO_2$ concentrations in the two depths during wind speeds above 3 m s$^{-1}$, there was a vertical gradient observed during wind still nights. The concentrations observed temporally ranged from undersaturation to supersaturation in 25 cm and 5 cm, respectively. Fluxes calculated from the measured concentrations therefore would change from negative to positive, depending on the measurement depth. Simultaneous Eddy Covariance measurements showed that even the measurements close to the surface underestimated the actual $CO_2$ concentration. Oxygen measurements support our hypothesis that respirational processes at the water surface cause a temporal $CO_2$ concentration gradient from the surface to the underlying water. Until now, the depth of $CO_2$ measurements has not been questioned, as long as measurements were done in the upper mixed layer and close to the surface. Our results provide evidence that representative measurements of $CO_2$ in the water strongly depend on depth and time of measurements.

## 1 Introduction

Lakes are hotspots of inland carbon cycling and are important sources of greenhouse gases (GHGs), such as carbon dioxide ($CO_2$). The significant role of $CO_2$ in global warming and climate dynamics makes quantifying its emission from various ecosystems, including lakes and reservoirs, important for developing strategies to combat climate change. Proper measurement of $CO_2$ emissions from lakes is the basis for robust global $CO_2$ emission quantification (Raymond et al., 2013; Lauerwald et al., 2023).

The state-of-the-art method to quantify GHG fluxes between earth surface and atmosphere is Eddy Covariance (EC) (Aubinet et al., 2012). Measurements by EC represent direct measurements covering large footprint areas at high temporal resolution. However, EC systems are expensive and complex to run. Thus, there are only few EC sites on lakes worldwide. In contrast to methane, which is often emitted by gas bubbles, $CO_2$ fluxes at the water surface are nearly exclusively driven by diffusion. The by far most used approach to quantify diffusive aquatic GHG fluxes is the Thin Boundary Layer approach (TBL, Lauerwald et al., 2023), also referred to as flux gradient method). The TBL approach, in contrast to EC, is an indirect measurement to quantify diffusive gas exchange across the water atmosphere interface. The TBL flux is derived from the



concentration gradient between the water and the atmosphere ($\Delta c_{CO_2}$) multiplied by the gas exchange velocity ($k$, equation 1).

$$F = \Delta c_{CO_2} * k, \qquad\qquad\qquad (1)$$

This method is much simpler than EC, because it only requires concentration measurements in both the water and

atmosphere. The gas exchange velocity can be estimated from meteorological data, typically wind speed and eventually temperature or fetch, using empirical models (Cole and Caraco, 1998; McGillis et al., 2004; MacIntyre et al., 2010; Vachon and Prairie, 2013; MacIntyre et al., 2021). The temporal resolution can be comparable to the EC method when using submerged $CO_2$ probes. However, for both methods, uncertainties arise from situations involving very low wind speeds.

When there is no wind, EC systems cannot measure at all, because the method relies on air movements (Aubinet et al., 2012;

Podgrajsek et al., 2014). But also, the TBL approach encounters difficulties under wind still conditions. While wind is the primary driver of gas transfer velocity and surface turbulence in large water bodies, its influence decreases under calm conditions, where factors such as surface cooling-induced convection (MacIntyre et al., 2010) and the lake's morphometry (Schilder et al., 2013) gain prominence in controlling gas exchange velocities. Consequently, predicting k based on wind speed alone introduces significant uncertainties. Thus, there is an urgent need to improve our ability to quantify aquatic GHG

fluxes under low turbulence conditions.

While there are several attempts to improve the models that predict $k$ (Cole et al., 2010; Crusius and Wanninkhof, 2003; MacIntyre et al., 2010; Vachon and Prairie, 2013), to our knowledge, the effect of uncertainty in $c_{CO_2 Water}$ measurements on the TBL approach has hardly been investigated. In calm conditions, the absence of wind significantly reduces turbulence, leading to surface microstratification. This microstratification creates distinct layers within the mixed layer, each with

varying temperatures and solute concentrations, including $CO_2$ (Åberg et al., 2010). Specifically, microstratification may result in dissolved gas gradients near the water's surface, suggesting that the actual layer engaging in atmospheric exchange becomes very thin. This challenges the assumption of the TBL approach that water just below the surface accurately represents the layer of gas exchange.

The critical layer of diffusive gas exchange is the surface micro layer (SML). The SML is the interface between the water

and the atmosphere and is characterized by high biological activity and physical processes that affect the interaction with the atmosphere (Gladyshev, 2002; Wurl et al., 2011). The SML is known for its enriched concentrations of algae, organic and inorganic solutes, and particles (Hardy, 1982). These components collectively distinguish the SML as a unique layer situated between the atmosphere and the underlying water. In the ocean, surfactants in the SML were found to reduce the diffusive gas exchange with the atmosphere by 32 % (Pereira et al., 2018). Under calm conditions, the SML thickens, becoming even

more crucial for the diffusive exchange of gases (Rahlff et al., 2017). However, despite its importance, there remains a gap in our understanding of the interactions between the atmosphere, the SML, and the water beneath, particularly in fresh water environments. Dynamics in gas exchange between the epilimnion, the surface layer and the atmosphere could lead to systematic uncertainty in the quantification of the surface gas concentration, as samples might be collected from depths that





do not accurately reflect the conditions of the water-atmosphere interface. In a recent study, Rudberg et al. (2024) explored
how spatial and temporal differences affect the influence of k and $c_{CO2}$ on the variability of $F_{CO2}$. By deploying a floating
chamber over several hours, the gas concentration in the chamber equilibrated with the gas concentration in the surface water
and enabled the quantification of the surface $CO_2$ concentration. With this approach they demonstrated that over long-term
periods, $c_{CO2}$ contributed more to $F_{CO2}$ variability than k. This finding emphasizes the need for precise $c_{CO2}$ measurements
when estimating fluxes using models. Similar research has been conducted in the ocean. Although $CO_2$ samples for gradient-
based flux models are usually collected a few meters below the water surface, Calleja et al. (2013) found significant
gradients between depths of 5-8 meters. Hari et al. (2008) found different $CO_2$ concentrations in 0.1 m and 0.5 m depth while
investigating a new method for $CO_2$ measurements. However, while these differences are attributed to varied biological
activities in the different depths, there is a lack of knowledge about the formation and characteristics of such gradients,
especially with regard to the SML and diffusive gas exchange with the atmosphere.
To better understand the importance of vertical $CO_2$ gradients at the water-atmosphere interface, we conducted an extensive
field experiment in a eutrophic reservoir. Our study is based on two key hypotheses: firstly, that temporal gradients of $CO_2$
concentrations are present close to the water surface, and secondly, that these gradients are influenced by meteorological
factors, such as wind. To assess the effect of an eventual surface $CO_2$ gradient on $CO_2$ fluxes we compared fluxes calculated
by the TBL approach with fluxes measured by EC during the same period.

## 2 Materials and methods

### 2.1 Study site

Measurements were made at Bautzen reservoir in Germany (51.218 °N, 14.466 °E). It is a dimictic reservoir, but high wind
exposure, a relatively large surface area (533 ha) as well as a circular shape, and the shallow depth (mean 7.4 m, maximum
13.5 m) result in a weak stratification during summer in some years (Benndorf, 1995; Spank et al., 2023). The eutrophic to
hypereutrophic reservoir (Kerimoglu and Rinke, 2013) serves for flow regulation of river Spree – the main river of Berlin. A
preceding study showed $CO_2$ uptake during the ice-free season of −9.8 and −71.0 g C m$^{-2}$ during the ice-free season of 2018
and 2019, and $CH_4$ fluxes of 24.0 g C m−2 and 23.2 g C m$^{-2}$, respectively (Spank et al., 2023).

### 2.2 Meteorological field observatory

A floating outdoor laboratory (FOL) was operated to continuously observe the mass- and energy exchange between the
water surface and the atmosphere. A detailed description of FOL and its instrumentation of can be found in Spank et al.
(2020, 2023, 2024). The FOL provided reference data of the carbon dioxide flux ($F_{CO2}$) between the water surface and the
atmosphere as well as data of wind speed (U), air temperature ($T_a$), relative air humidity (RH), air pressure ($p_a$), solar
radiation ($R_g$) and water temperature ($T_w$) in a temporal resolution of 30 minutes. In addition, the FOL served as a carrier for
the devices used during the measurement campaigns. The measurement height of meteorological sensors was 2 m above the



water surface in accordance with international standards. The thermistor chain used was configured to measure $T_w$ at depths
of 0.25, 0.50, 0.75, 1.0, 1.5, 2.0, 2.5, 3.0, 4.0, 5.0, 6.0, 8.0, 10.0, 12.5, and 15.0 m. The eddy covariance measuring system,
which provides $F_{CO_2}$, was instrumented according to standards and guidelines given by Lee et al. (2004), Foken and Mauder
(2024), Aubinet et al. (2012) and Burba (2013). The EC data are representative for the pelagic zone of Bautzen reservoir. In
particular, carefully performed footprint analyses proved that effects and disturbances from surrounding terrestrial sites can

be almost completely ruled out (Spank et al. 2023). The EC post processing was based on the methodologies of Carbo
Europe (Aubinet et al., 1999) and ICOS (Sabbatini et al., 2018) and had been performed utilizing the software EddyPro 7.0.8
(LI-COR Biosciences 2023). However, the special measurement condition on a floating platform had to be taken into
accounted, which required an upstream correction of the sensor misalignment (Spank et al., 2020, 2023).

### 2.3 Dissolved Gases

To detect eventually occurring dissolved $CO_2$ concentration gradients at the water surface, we deployed two $CO_2$ probes
(Contros HydroC, -4H- JENA engineering, Germany) at different depth from September 18 to September 25 in 2022
(hereinafter referred to as *study period*). These probes, featuring diffusive membranes with a diameter of 8 cm, were
positioned so that the central points of their membranes were situated at depths of 0.05 m and 0.25 m, respectively. We used
a frame that consisted of two aluminium bars, which were connected by two cross bars in the middle. Aluminium extensions

on the cross bars were used to mount the $CO_2$ probes horizontally and allow adjustments of the measurement depths.
Buoyancy floats in the four corners were used to make the frame float below the surface (Figure S 1). The $CO_2$ probes were
powered by 24 VDC provided by the floating platform. To maintain the integrity of surface water stratification, we left the
membranes of the $CO_2$ probes uncovered. $CO_2$ measurements were logged internally by the probes every minute, with a
measurement cycle of 1400 minutes measuring, followed by 3 minutes zeroing and 37 minutes flushing. $CO_2$ probe

performance was validated before deployment by running both probes in a sink, using an infrared gas analyser (EGM-5, PP-
Systems, USA) coupled to a membrane contactor (MiniModule, Liqui-Cel, USA) as a reference system (as in Koschorreck et
al., 2021; Figure S 2). After deployment, internal data required post processing (as described in Fietzek et al., 2014) because
the measurements were out of the factory calibration range. In brief, the internal zero measurements using $CO_2$ absorbents
generate frequent calibration points for $pCO_2 = 0$. Daily zero measurements were used to calibrate signal outputs between

the last and the next zero measurement. Finally, a modified polynomial function was used to determine corrected $CO_2$
concentrations. During our sampling period, the reference signal was constant, indicating proper function of the sensor.
Oxygen concentrations were measured every 15 minutes by optical $O_2$ loggers. Surface $O_2$ concentration at 0.05 m depth
was measured with a miniDOT logger equipped with a miniWIPER (Precision Measurement Engineering, USA, wiping
frequency 12 hours) mounted at the same depth as the surface $CO_2$ probe. A D-Opto oxygen logger (ZebraTech, Nelson, NZ)

was installed in the bottom water at 1 m above the sediment using a separate mooring. Both oxygen sensors were calibrated
using a two-point calibration at 0 % and 100 % oxygen saturation and corrected for potential drift.



## 2.4 Data analysis and Statistics

The platform data was prepared and exported from Python. For this study, the platform data, $CO_2$ probe data and oxygen data were compiled in R (R Core Team, 2023). All subsequent analyses, statistical computations, and visualizations were
performed in R. $CO_2$ and oxygen measurements were averaged over 30-minute periods. Day and night averages were calculated using sunrise and sunset times determined using the *is.day()* function. Gas transfer velocities were calculated from wind speeds at 10 meter height , using the k.cole.base() function, referring to the parametrization explained in Cole and Caraco (1998). Gross primary production (GPP) and respiration (R) were calculated using the metab() function. Those functions are part of the LakeMetabolizer package (version 1.5.5; Winslow et al., 2016). Gas fluxes were calculated from in
situ water and air $p_{CO2}$ and $k$ using equation 1.

## 3 Results

In 2022, Bautzen reservoir was thermally stratified from the beginning of May. During the stratification period (mean mixed layer depth = 4.4 m), the maximum temperature at the surface was 30 °C, while the temperature above the ground reached a maximum of 13 °C (Figure A 1). On September 9, 2022, a thunderstorm with strong winds hit the Bautzen region, leading to
a shutdown and subsequent 5-day outage of the measurement platform. This storm also marked the end of the stratified season and started mixing of the reservoir. On September 18[th], which marks the beginning of our extensive GHG measurements, $T_w$ was 16 °C at both the surface and bottom. The oxygen $C_{O2}$ was 9.1 mg L[-1] at both the surface and the bottom.







**Figure 1: CO₂ concentrations measured in the two depths (a), Wind speed (b), Water temperature at 0.25 m depth + air temperature (c), incoming short-wave radiation (SW, d), and oxygen (e). Horizontal bars show mean values of days and nights. Grey highlights show night-time.**





The $CO_2$ concentration at the water surface showed a consistent diurnal pattern during the entire study period. At night, $CO_2$
concentration at 5 cm depth was generally higher than at 25 cm depth. Thus, every night of our study period a gradient of
$CO_2$ concentration near the water surface developed. In contrast, no such gradient was observed during the. At both depths,
$CO_2$ concentrations increased with the disappearance of short-wave radiation at sunset and decreased with its increase at
sunrise (Figure 2d). Mean air temperatures ($T_a$) ranged between 9 °C and 15 °C, with distinct diurnal patterns. The water
temperature at 25 cm depth ($T_{w25}$) decreased slightly from 16 °C to 15 °C during the measurement period, except for
September 23rd, 24th, and 25th, when water temperature increased by 1 °C during the day (Figure 1 C). During the days of
September 23rd, 24th, and 25th, the air temperature was above the water temperature. On all other days and nights, the water
temperature consistently remained higher than the air temperature.

While the diurnal pattern of the $CO_2$ gradient at the water surface was consistent during our study period we observed large
differences regarding the magnitude of this gradient. Our measurement period can be divided into two parts with differing
weather conditions. The first period from September 19th to 21st was windy with U mostly above 3 m s$^{-1}$. However, during
the second half of our sampling period, wind speeds were mostly below 3 m s$^{-1}$, with 63 % of those times even falling below
1 m s$^{-1}$ (Figure 1 b). The $CO_2$ concentrations near the water surface during the night showed a fundamentally different
behaviour during those two periods (Figure 1 a). High $CO_2$ concentrations up to 125 µmol L-1 were observed during the
calm period while during the windy period $CO_2$ was permanently undersaturated and at the detection limit of our probes.
175   Notably, nightly $CO_2$ concentrations in both depths were ten to twenty times higher during the calm period compared to the
windy period. Starting on September 22nd, nightly $CO_2$ concentrations exceeded the equilibration concentration, leading to
temporary supersaturation in calm nights. While the nightly mean $CO_2$ concentration at 5 cm depth exceeded the
equilibration concentration in the nights of September 23rd to 26th, the mean concentration at 25 cm depth never surpassed
equilibrium. The oxygen concentration was different between the two periods. $O_2$ concentrations at both 5 cm depth and at
180   the bottom were slightly undersaturated during the windy period. When the water column started stratifying again (Figure
1 e), $O_2$ concentrations at the two depths started to diverge. While the $O_2$ concentration at the bottom continuously decreased
over the rest of the sampling period, the concentration at 5 cm increased to supersaturation, showing clear diurnal patterns of
oxygen production and consumption (Figure 1 e).





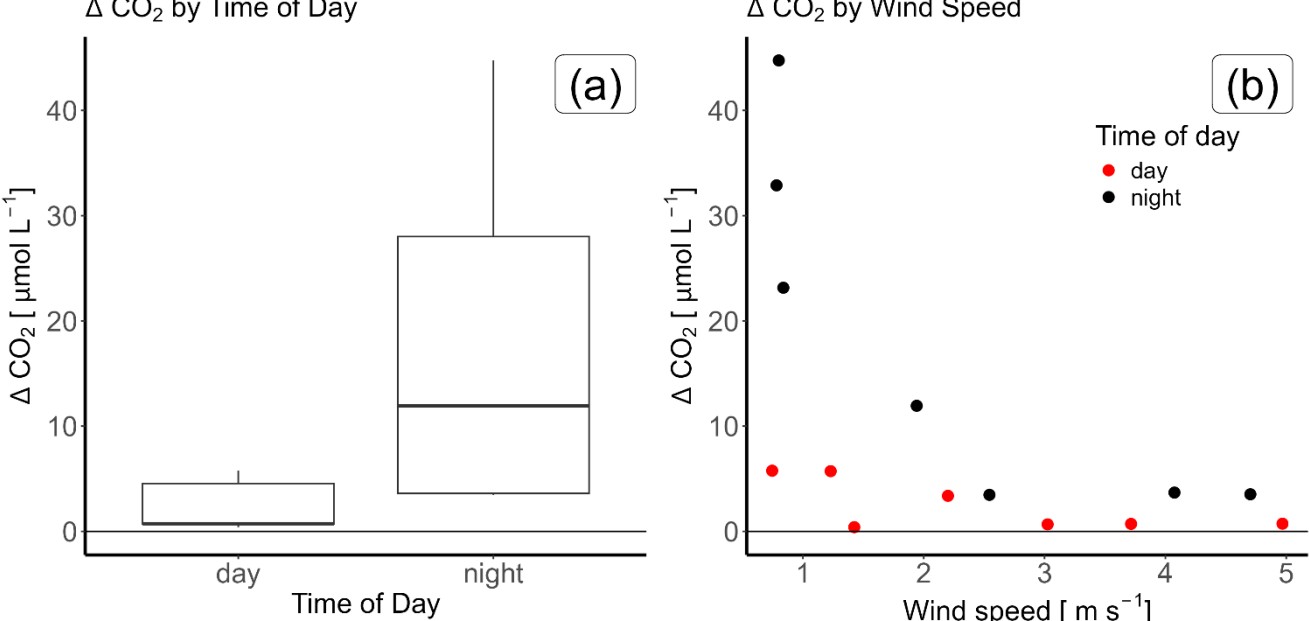

**Figure 2: Dependence of CO2 difference between 5 cm and 25 cm depth on time of day (a) and wind speed (b). Both plots are based on mean values per time of day.**

To gain a more detailed understanding of the $CO_2$ gradients at the surface, we calculated the difference in $CO_2$ concentrations between 5 cm and 25 cm depths. Our measurements showed that the mean concentration gradient was significantly steeper during the night compared to during the day (t-test, $p < 0.05$, Figure 2 a). The magnitude of the gradient depended on wind speed (Figure 2 a). Generally, the mean $CO_2$ gradient was lower at high wind speeds for both day and night. However, while low wind speeds did not lead to higher $CO_2$ gradients during the day, the gradients in calm nights were 5 to almost 10-fold steeper (Figure 2 b). This diurnal development of a $CO_2$ gradient at the surface should have affected $CO_2$ emissions.





**Figure 3: EC TBL comparison. Fluxes calculated from TBL approach (a) and measured by Eddy Covariance (b) with hourly resolution as well as averaged over every day and night periods (c). Grey highlights show night-time.**

We used our $CO_2$ concentration data to calculate $CO_2$ fluxes by the TBL approach and compared those fluxes with measurements done with Eddy Covariance (Figure 3). During the windy period, TBL fluxes were negative regardless of which $CO_2$ probe data we used and comparable to the fluxes measured by EC (Figure 3 a, b, and c). In calm nights, supersaturation of $CO_2$ led to positive TBL fluxes with data from both depths. Because at 25 cm depth oversaturation was only reached at the end of the night, mean night fluxes derived from 25 cm depth remained negative (Figure 3 c). EC fluxes during those nights were positive too, but significantly higher than TBL fluxes (Figure 3 b, c). In 5 out of 14 cases, the flux direction differed between the TBL approach and EC measurements, when using the measurements of the 25 cm probe for the TBL approach. In contrast, when using $CO_2$ measurements from the 5 cm probe, only 2 out of 14 instances show different flux directions. During our study period of 7 days, the total $CO_2$ emissions were -2.2 g m$^{-2}$ when calculated using the 5 cm data, -4.2 g m$^{-2}$ based on 25 cm data, and 4.6 g m$^{-2}$ as recorded by the eddy covariance system.



## 4 Discussion

Our data, showing the change from windy to calm weather conditions, and a mixed water column to formation of stratification, respectively, within a week, provided unique insights in the influence of microstratification on greenhouse gas
exchange between lakes and the atmosphere. Our detailed $CO_2$ concentration measurements clearly showed the temporal development of a $CO_2$ gradient at the water surface. This observation is consistent with previous studies like Hari et al. (2008), whose results indicate $CO_2$ gradients within the upper 50 cm of the water layer.

During the windy period, $CO_2$ concentrations were as low as the detection limit of our probes at daytime, probably caused by photosynthetic $CO_2$ consumption by abundant phytoplankton. $CO_2$ undersaturation in high productive lakes is a common
observation (Balmer and Downing, 2011; Zagarese et al., 2021). In that period, $CO_2$ concentrations were consistently lower than the atmospheric equilibrium concentration, turning the reservoir to a $CO_2$ sink during that time. The oxygen concentrations measured at the surface and above the bottom as well as the temperature profiles suggest that the entire water column was mixed in that period, likely leading to homogeneous $CO_2$ concentration across the whole water column, too. This observation is supported by the fact that no $CO_2$ gradient was measured during the day, while a slight gradient was
detected at night. This occurs because $CO_2$ is consumed quickly during daylight hours, whereas at night, the uptake of $CO_2$ from the atmosphere leads to slightly higher concentrations at the water surface. This was reflected by our 5 cm probe measuring concentrations closer to atmospheric equilibrium than the 25 cm probe.

When the wind ceased at September 21st, the vertical distribution of gas concentrations changed. Starting on that day, oxygen concentrations at the surface and at the bottom began to increase and decrease, respectively. This observation
indicates that the reservoir slowly underwent stratification again. While there was no difference in $CO_2$ gradients during calm daylight hours compared to windy days, we observed notably high $CO_2$ concentrations within the top 25 centimetres during calm nights. The high CO2 concentrations during calm nights could be related to the effects of stratification. During windy periods, the mixed water layer depth was greater than during calm periods, resulting in a larger volume of water influenced by O2 and CO2 overturn in the photo-active layer. Phytoplankton may have decreased the $CO_2$ concentration in
this large volume through photosynthesis over the days of strong winds. At night, respiration slightly increased the $CO_2$ concentration, but this was not enough to fully compensate for the daytime reduction. When windspeed decreased, the mixing depth and volume became smaller. During these calmer nights, respiration in the shallow mixed layer led to significant $CO_2$ accumulation at the surface. Additionally, microstratification within the top 25 cm of the water column could further restrict the volume available for $CO_2$ accumulation, potentially leading to even higher concentrations (Figure 4). Our
findings are consistent with a study of Åberg et al. (2010), who found that short term CO2 variations at the surface were best related to thermal dynamics within the upper mixed layer, whereas parameters like wind and radiation did not influence CO2 concentrations. In our study we found negative effect of incoming solar radiation on CO2 concentrations during the day, but a positive feedback during the night. Further, varying CO2 concentrations haven been related to changes in lake metabolism after storms before (Vachon and del Giorgio, 2014).





Interestingly, the 5 cm probe, despite being closer to the surface, recorded higher $CO_2$ concentrations than the underlying water. This requires a $CO_2$ producing process in the surface layer, likely biological respiration. There are various groups of organisms that could increase $CO_2$ in the 5 cm layer, such as neuston, which comprises organisms living at or even within the surface micro layer. Phytoplankton was found to float at the water surface at wind speeds lower than 3 m s$^{-1}$ (Zhang et al., 2021). Further, some species migrate from the lower boundary of the epilimnion to the surface during night. This behaviour

is called diel vertical migration and is used to access food while avoiding predators that need light for hunting (Ringelberg, 1999). Both neuston and migrating species are respiring during the night, thereby producing $CO_2$. Furthermore, the vertical mixing during windy periods likely increased nutrient availability for phytoplankton, potentially triggering growth and accumulation of algae at the water surface (Nürnberg et al., 2003). These phytoplankton switch from net photosynthesis during the day to respiration at night, thereby increasing $CO_2$ concentration at night. Mean gross primary production and

respiration in the calm period were 10.4 mg $O_2$ L$^{-1}$ d$^{-1}$ and -9.0 mg $O_2$ L$^{-1}$ d$^{-1}$, respectively, which is comparable to other hyper-eutrophic lakes, especially after storm events (Williamson et al., 2021). During algal blooms, the water surface is often covered by a mat of floating algae. Although we lack chlorophyll data, satellite images taken before and after our measurement period suggest rapid algal growth following the storm events that occurred just before our study (Figure B 1). Such mats can become very dense, possibly increasing $CO_2$ production and accumulation even more.

Moreover, the effect of convective mixing within the top water layer plays an important role in the distribution of $CO_2$ concentrations. Convective mixing, driven by water density differences due to cooling at the water surface, can enhance the vertical movement of water, thereby influencing the distribution of $CO_2$ in the water column. While we did not find signs of thermal microstratification at the surface during nights with strong $CO_2$ gradients, we contrastingly observed conditions favoring convection (Figure 1 c). Further, meteorological parameters such as atmospheric stability, emitted long wave

radiation, or sensible heat flux, indicate unstable conditions in the air above the water. From this, we infer that the biomass accumulated at the water surface could either produce $CO_2$ at rates that exceed its transport in the upper water layer, or the algal mats acted as a physical barrier between the water and air, leading to increased stability of the water beneath the mats by preventing atmospheric instability from influencing the water below.

The $CO_2$ gradient measured during calm nights was fundamentally influencing the fluxes calculated with the TBL method.

While it is generally acknowledged that daytime measurements do not reflect the concentrations at night (Erkkilä et al., 2018), the depth of $CO_2$ measurements has not been questioned so far, as long as measurements were done in the upper mixed layer and close to the surface. Our measurements provide evidence that representative measurements of the $CO_2$ concentration in the water strongly depend on depth and time of measurements. In our results, the depth of measurement even determined whether the TBL method would result in efflux or influx (Figure 3bn). This was especially visible at night

when the flux calculated from the $CO_2$ concentration measured at 25 cm depth was negative, while both EC measurements and TBL based on the measurements in 5 cm showed significant positive fluxes. The fact that the EC fluxes were higher than the TBL fluxes based on the 5 cm probe could be explained by the $CO_2$ gradient which probably continued towards the water-atmosphere interface. Our uppermost probe measured the mean $CO_2$ concentration between 1 and 10 cm depth and it





is plausible that the concentration at the water surface was even higher than measured by that probe. To measure the "real"
concentration at the surface $CO_2$ measurements would need to be conducted even closer to the SML. Floating chambers have
been used to measure $CO_2$ in the surface by deploying them as closed systems, therefore equilibrating the chamber volume
with the surface water (Rudberg et al., 2021). However, equilibration for that method is slow and introduces a time delay for
measured concentrations, which means that the temporal resolution is weak. Therefore, the method is not suited to directly
measure $CO_2$ concentration on short time scales and with high vertical resolution.


**Figure 4: Schematic of $CO_2$ accumulation in the surface water, with $CO_2$ probes located in 5 cm and 25 cm depth. (a): wind-induced turbulence causes homogeneous distribution of phytoplankton, resulting in $CO_2$ decrease in both measuring depths during the day. (b): wind induced turbulence causes homogeneous distribution of phytoplankton. $CO_2$ is produced by respiration, but the $CO_2$ concentration across the whole water column is low, thus respiration is compensating strong undersaturation. $CO_2$**
**uptake from the atmosphere causes slightly increased concentration in 5 cm. (c): $CO_2$ concentration is low in the whole mixed**



layer, but calm conditions cause phytoplankton to accumulate at the surface. (d): Calm conditions cause phytoplankton to accumulate at the water surface. Respiration causes $CO_2$ increase in this layer, which is causing $CO_2$ emissions during calm nights.

**Conclusions**

Our results show that the surface $CO_2$ gradient is regulated by an interplay of physical and biological processes (Figure 4).
Only when limited mixing of the surface layer comes together with high accumulation of biomass in the surface water a thin surface layer with high $CO_2$ concentrations can develop. That effect can turn a $CO_2$ under-saturated lake to a temporary $CO_2$ source.

**Appendix A: Detailed water temperature graphs**

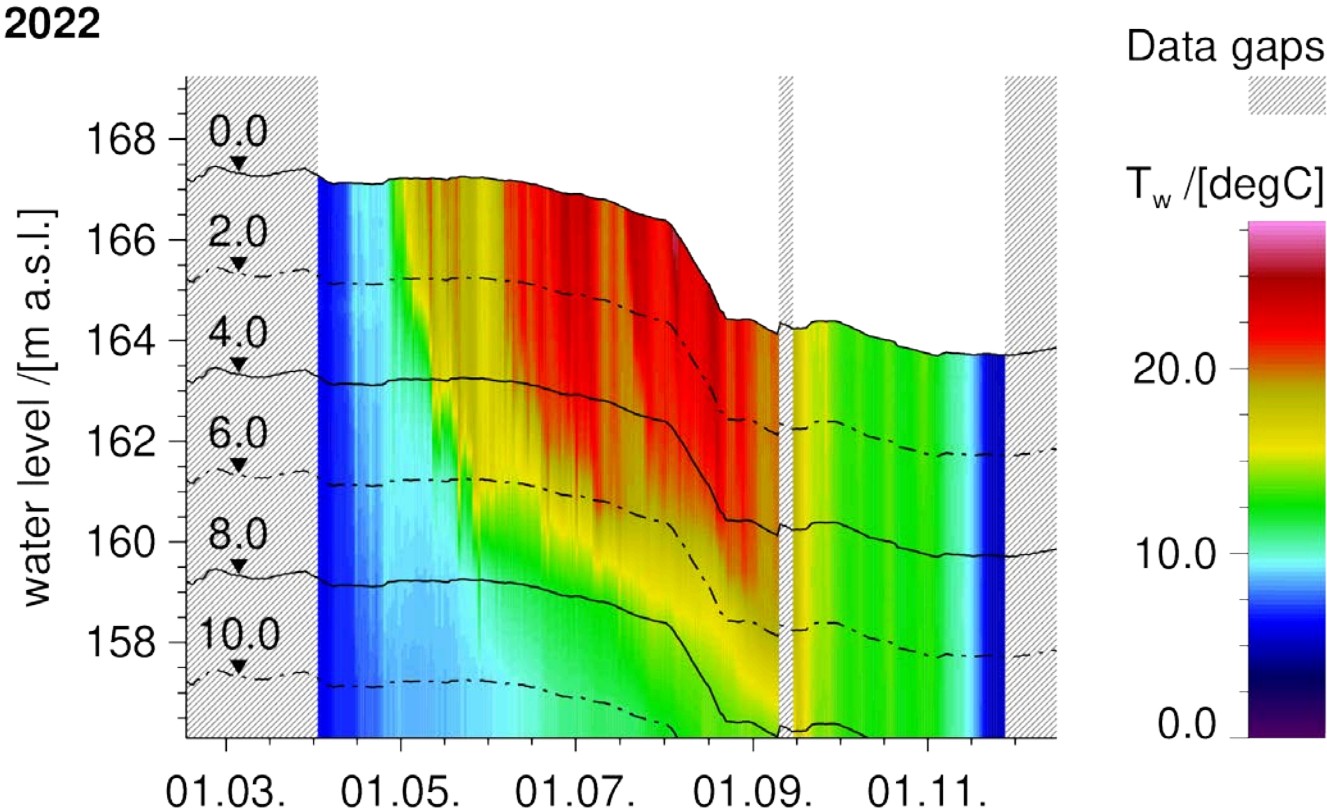

**Figure A 1: Water level (meters above sea level) and water temperature profile from April 1st to November 28th in 2022 at Bautzen reservoir. Solid and dot-dashed black lines indicate water depths of 2 m, 4 m, 6 m, 8 m, and 10 m. Grey bars highlight missing data.**





**Figure A 2: Detailed water temperatures across the water column during the extensive sampling period. Dotted lines highlight surface and bottom temperatures. Grey highlights show night-time.**



**Appendix B: Satellite images of the reservoir**

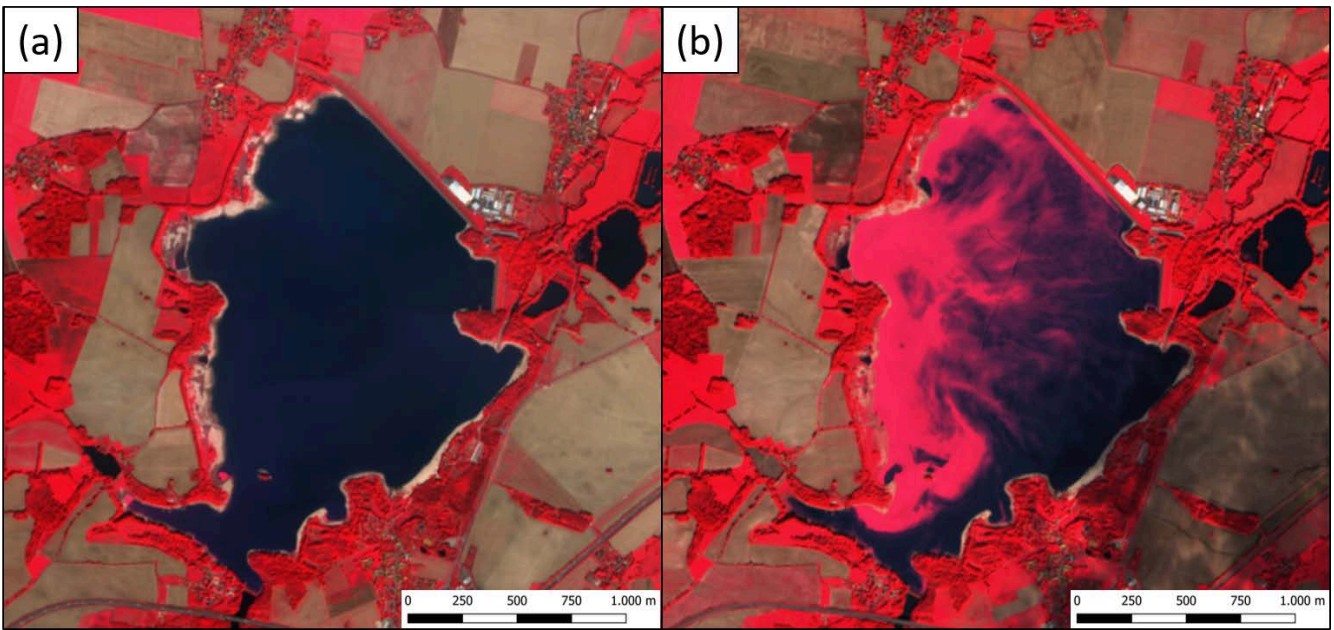

**Figure B 1: False colour images from Sentinel-2 L2A taken on September 5th (a) and 25th (b), 2022, provided by the European Space Agency (ESA) and accessed via the EO Browser (https://apps.sentinel-hub.com/eo-browser/, Sinergise Solutions d.o.o., a Planet Labs company). September 5th was selected because it was the last day without cloud coverage before the sampling period. The images show the state of the reservoir during the algal bloom described in this manuscript. The false colour images, which are typically used to highlight specific features like vegetation or water, do not show the water surface properly on September 25th, due to algal coverage. This clearly highlights the intensity of the algal bloom, which was likely covering the water surface.**


**Code availability**

Code is available upon request to the corresponding author.

**Data availability**

Data are available upon request to the corresponding author.

**Author contribution**

All authors contributed to the study's conception and design. Field measurements were carried out by PA and US. Meteorological data were processed by US, while limnological data were processed by PA. PA performed the data analysis. PA wrote the first draft of the manuscript, and all authors contributed to the final version. All authors read and approved the final manuscript.





**Competing interests0**

The authors declare that they have no conflict of interest.

**Acknowledgements**

We thank Martin Wieprecht for his help during the field campaign. We further thank Muhammed Shikhani for valuable advice on data analysis and Peifang Leng for her insightful comments on the manuscript. This study was funded by the German Science Foundation (Deutsche Forschungsgesellschaft, DFG) in the frame of the project "Meteorological Drivers of

Mass and Energy Exchange between Inland Waters and the Atmosphere – MEDIWA" (project number: 445326344).

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
