# Peer review of "Surface CO2 Gradients Challenge Conventional CO2 Emission Quantification in Lentic Water Bodies under Calm Conditions."

_EGUsphere, 2024_

## Referee Comment (RC1)

[revised manuscript text omitted]

check how to cite that correctly, at least the version needs to be stated I think
data were compiled in R (R Core Team, 2023). All subsequent analyses, statistical computations, and visualizations were

140    performed in R. $CO_2$ and oxygen measurements were averaged over 30-minute periods. Day and night averages were

calculated using sunrise and sunset times determined using the *is.day()* function. Gas transfer velocities were calculated from

wind speeds at 10 meter height , using the k.cole.base() function, referring to the parametrization explained in Cole and
How do you transform your 2-m wind to 10-m?
Caraco (1998). Gross primary production (GPP) and respiration (R) were calculated using the metab() function. Those

functions are part of the LakeMetabolizer package (version 1.5.5; Winslow et al., 2016). Gas fluxes were calculated from in

145    situ water and air $p_{CO2}$ and *k* using equation 1.

Probably nice to know the "maths" behind the function for GPP and R, for example

**3 Results**

In 2022, Bautzen reservoir was thermally stratified from the beginning of May. During the stratification period (mean mixed
layer depth = 4.4 m), the maximum temperature at the surface was 30 °C, while the temperature above the ground reached a
maximum of 13 °C (Figure A 1). On September 9, 2022, a thunderstorm with strong winds hit the Bautzen region, leading to

150    a shutdown and subsequent 5-day outage of the measurement platform. This storm also marked the end of the stratified
season and started mixing of the reservoir. On September 18[th], which marks the beginning of our extensive GHG
measurements, $T_w$ was 16 °C at both the surface and bottom. The oxygen $C_{O2}$ was 9.1 mg L[-1] at both the surface and the
bottom.

Out of curiosity (and also for paragraph consistency), do you have measurement of oxygen before the mixing,
when temperature was 30°C?

The result section needs a bit of re-estructure for clarity. For example, you talk about CO2 during the day in line 159 and again in line 17
I suggest to unify. Same get for other parameters.

[Figure]

[Figure]

add in M&M how this equilibrium is calculated. I guess is the CO2 from atmosphere (?). How did you get it?

[Figure]

consistency: in the main text you use Ta and Tw. Also, do you have the temperature at 5cm?

and at 25 cm?

**Figure 1: CO₂ concentrations measured in the two depths (a), Wind speed (b), Water temperature at 0.25 m depth + air temperature (c), incoming short-wave radiation (SW, d), and oxygen (e). Horizontal bars show mean values of days and nights. Grey highlights show night-time.**

dissolved

the bars are a bit misliding, as sometimes there is a lot of variation. I was wondering if it could make more sense to add a box plot, where you will also show the mean and the range

[Figure]

The $CO_2$ concentration at the water surface showed a consistent diurnal pattern during the entire study period. At night, $CO_2$
160    concentration at 5 cm depth was generally higher than at 25 cm depth. Thus, every night of our study period a gradient of
$CO_2$ concentration near the water surface developed. In contrast, no such gradient was observed during the. At both depths,
$CO_2$ concentrations increased with the disappearance of short-wave radiation at sunset and decreased with its increase at
sunrise (Figure 2d). Mean air temperatures ($T_a$) ranged between 9 °C and 15 °C, with distinct diurnal patterns. The water
temperature at 25 cm depth ($T_{w25}$) decreased slightly from 16 °C to 15 °C during the measurement period, except for
165    September 23rd, 24th, and 25th, when water temperature increased by 1 °C during the day (Figure 1 C). During the days of
September 23rd, 24th, and 25th, the air temperature was above the water temperature. On all other days and nights, the water
temperature consistently remained higher than the air temperature.

While the diurnal pattern of the $CO_2$ gradient at the water surface was consistent during our study period we observed large
differences regarding the magnitude of this gradient. Our measurement period can be divided into two parts with differing
U 10 m or U 2m? Also mentioned in the figure
170    weather conditions. The first period from September 19th to 21st was windy with U mostly above 3 m s$^{-1}$. However, during
the second half of our sampling period, wind speeds were mostly below 3 m s$^{-1}$ with 63 % of those times even falling below
Unify how you reference the figure, sometime a or A, sometimes space or not between number and letter
1 m s$^{-1}$ (Figure 1 b). The $CO_2$ concentrations near the water surface during the night showed a fundamentally different
behaviour during those two periods (Figure 1 a). High $CO_2$ concentrations up to 125 µmol L-1 were observed during the
which is? needs to be stated in the M&M
calm period while during the windy period $CO_2$ was permanently undersaturated and at the detection limit of our probes.
175    Notably, nightly $CO_2$ concentrations in both depths were ten to twenty times higher during the calm period compared to the
windy period. Starting on September 22nd, nightly $CO_2$ concentrations exceeded the equilibration concentration, leading to
temporary supersaturation in calm nights. While the nightly mean $CO_2$ concentration at 5 cm depth exceeded the
equilibration concentration in the nights of September 23rd to 26th, the mean concentration at 25 cm depth never surpassed
equilibrium. The oxygen concentration was different between the two periods. $O_2$ concentrations at both 5 cm depth and at
180    the bottom were slightly undersaturated during the windy period. When the water column started stratifying again (Figure
1 e), $O_2$ concentrations at the two depths started to diverge. While the $O_2$ concentration at the bottom continuously decreased
over the rest of the sampling period, the concentration at 5 cm increased to supersaturation, showing clear diurnal patterns of
oxygen production and consumption (Figure 1 e).

I guess the answer is no but do you have measurements of CO2 in the bottom? If not, worth to considered for the next project :)

[Figure]

Carefull on using deltaCO2 for two different things: air-water (as in Equation 1) or 5-25 cm.
Find another abbreviation of metion all the time deltaCO2_(5-25cm)

[Figure]

**Figure 2: Dependence of CO2 difference between 5 cm and 25 cm depth on time of day (a) and wind speed (b). Both plots are based on mean values per time of day.** See Stolle et al. (2020), figure 13. We presented k but could have also a relationship.

To gain a more detailed understanding of the $CO_2$ gradients at the surface, we calculated the difference in $CO_2$ concentrations between 5 cm and 25 cm depths. Our measurements showed that the mean concentration gradient was significantly steeper during the night compared to during the day (t-test, $p < 0.05$, Figure 2 a). The magnitude of the gradient depended on wind speed (Figure 2 a). Generally, the mean $CO_2$ gradient was lower at high wind speeds for both day and night. However, while low wind speeds did not lead to higher $CO_2$ gradients during the day, the gradients in calm nights were 5 to almost 10-fold steeper (Figure 2 b). This diurnal development of a $CO_2$ gradient at the surface should have affected $CO_2$ emissions.

[Figure]

[Figure]

**Figure 3: EC TBL comparison. Fluxes calculated from TBL approach (a) and measured by Eddy Covariance (b) with hourly resolution as well as averaged over every day and night periods (c). Grey highlights show night-time.**

It is always nicer when you can understand the caption and figure without the need to read the paper. Please explain all more precise

We used our $CO_2$ concentration data to calculate $CO_2$ fluxes by the TBL approach and compared those fluxes with careful with the wording, 5 ms-1 is not particularly windy :)

measurements done with Eddy Covariance (Figure 3). During the windy period, TBL fluxes were negative regardless of which $CO_2$ probe data we used and comparable to the fluxes measured by EC (Figure 3 a, b, and c). In calm nights,

200 supersaturation of $CO_2$ led to positive TBL fluxes with data from both depths. Because at 25 cm depth oversaturation was only reached at the end of the night, mean night fluxes derived from 25 cm depth remained negative (Figure 3 c). EC fluxes during those nights were positive too, but significantly higher than TBL fluxes (Figure 3 b, c). In 5 out of 14 cases, the flux direction differed between the TBL approach and EC measurements, when using the measurements of the 25 cm probe for the TBL approach. In contrast, when using $CO_2$ measurements from the 5 cm probe, only 2 out of 14 instances show emission through your study period? We need a time so you can get rid of your time unit.

205 different flux directions. During our study period of 7 days, the total $CO_2$ emissions were -2.2 g m$^{-2}$ when calculated using the 5 cm data, -4.2 g m$^{-2}$ based on 25 cm data, and 4.6 g m$^{-2}$ as recorded by the eddy covariance system.

How relieable are the EC-flux without wind/very low wind speed? You mentioned before they have problems.

[Figure]

The discussion is again very interesting but difficult to read. I suggest to find the main finding of each figure, explaining in the result section and then discussed here. If the journal allow, adding headers sections to the discussion will also help the reader

**4 Discussion**

[revised manuscript text omitted]

During day, there is also a phytoplankton CO2 release, just that problably the uptake is bigger.
Also just noticing, sometimes you talk about phytoplankton and sometime algae. In oceanography we use phytoplankton

[Figure]

[Figure]

**You are missing here the following explanation, to be consistent like in (d)**

**layer, but calm conditions cause phytoplankton to accumulate at the surface. (d): Calm conditions cause phytoplankton to accumulate at the water surface. Respiration causes $CO_2$ increase in this layer, which is causing $CO_2$ emissions during calm nights.**

**Conclusions**

Our results show that the surface $CO_2$ gradient is regulated by an interplay of physical and biological processes (Figure 4).

290  Only when limited mixing of the surface layer comes together with high accumulation of biomass in the surface water a thin

surface layer with high $CO_2$ concentrations can develop. That effect can turn a $CO_2$ under-saturated lake to a temporary $CO_2$

source.     **very short conclusion section. Here you should emphasize what is your study implication**

**Appendix A: Detailed water temperature graphs**

[Figure]

295  **Figure A 1: Water level (meters above sea level) and water temperature profile from April 1st to November 28th in 2022 at Bautzen reservoir. Solid and dot-dashed black lines indicate water depths of 2 m, 4 m, 6 m, 8 m, and 10 m. Grey bars highlight missing data.**

[Figure]

[Figure]

**Figure A 2: Detailed water temperatures across the water column during the extensive sampling period. Dotted lines highlight surface and bottom temperatures. Grey highlights show night-time.**

It is not surface, it is 25 cm :)

[Figure]

**Appendix B: Satellite images of the reservoir**

[Figure]

I would say here what the false colour shows. We need to read quite a lot to find it out.

**Figure B 1: False colour images from Sentinel-2 L2A taken on September 5th (a) and 25th (b), 2022, provided by the European**
305 **Space Agency (ESA) and accessed via the EO Browser (https://apps.sentinel-hub.com/eo-browser/, Sinergise Solutions d.o.o., a Planet Labs company). September 5th was selected because it was the last day without cloud coverage before the sampling period. The images show the state of the reservoir during the algal bloom described in this manuscript. The false colour images, which are typically used to highlight specific features like vegetation or water, do not show the water surface properly on September 25th, due to algal coverage. This clearly highlights the intensity of the algal bloom, which was likely covering the water surface.**

If you get a phytoplankton bloom, it could probably also be affecting temperature in the SML.
See Wurl et al. 2018 ( doi.org/10.1029/2018GL077946)

310 **Code availability**

Code is available upon request to the corresponding author.

**Data availability**

Data are available upon request to the corresponding author.

Is this allowed by the journal and DFG? I will strongly recommend/suggest to submit to PANGAEA or similar repository.

[revised manuscript text omitted]

---

## Author Comment (AC1)

Surface CO2 Gradients Challenge Conventional CO2 Emission Quantification in Lentic Water Bodies under Calm Conditions
Patrick Aurich
en

The manuscript titled "Surface $CO_2$ Gradients Challenge Conventional $CO_2$ Emission Quantification in Lentic Water Bodies under Calm Conditions" by Patrick Aurich et al. presents an insightful study on $CO_2$ dynamics in the near-surface layer of the Bautzen Reservoir. I think the first author is a PhD student, and I would like to extend my congratulations to him, recognizing the complexity and difficulty of this research.

The manuscript benefits from a high temporal and spatial resolution dataset, and the analysis provides valuable insights worthy of publication. However, I think the manuscript requires further development and refinement before it is ready for publication. As I am not concerned with remaining anonymous (and I can't remain anonymous given the nature of my comments 😊), I have attached my detailed feedback in the accompanying PDF document.

I look forward to reviewing the revised version of this manuscript.

Response: Dear Mariana, thank you for the nice words and your suggestions on our manuscript. We are happy that you recognize the value of our study and that you think it is worth publishing in Biogeosciences. Please find our response to your comments below. In a revised version of the manuscript, we will address your suggestions for improvement and adjust the language, spelling, and style according to your feedback.

**Detailed comments**

Line 14: not defined yet, I assume thin boundary layer. But note that this abbreviation is normally used a thermal boundary layer

Response: Yes, we will add the abbreviation to the previous sentence.

Line 27: I would go a bit further on the implications, what does mean of global carbon cycle?

Response: Thank you for the comment. The implantations for the global carbon cycle are, to this point, unknown. The chances to find the conditions as found in our dataset are rare, however, the magnitude of fluxes speaks for itself. For this abstract, however, we think it is enough to highlight the existence of pronounced CO2 gradients, as this guides researchers in further studies to take the effect into account.

Line 40: You mention before earth surface and atmosphere and you suddenly change to water atmosphere. Also I think water-atmosphere, as we use air-sea

Response: Thank you for this comment. For clarity, we will change it from earth surface to water surface.

Line 55: also the fact that most of these k-parametrization assume a zero-intercept

Response: True. That's why we prefer to use a parametrization by Wanninkhof with a non-zero intercept. This aspect will be added to a revised version of the manuscript.

Line 69: check also Mustaffa et al., 2020 :)

Response: Thank you for your comment. Mustaffa et al 2020 is indeed a very good reference to highlight the global relevance of the effect. We will add the reference.

Line 72: some reader might not know what that is

Response: Thank you for this note. However, we think that the term epilimnion is textbook knowledge for limnologists and does not need further explanation.

**Line 74: here you use the -, unify throught the ms**

Response: Thank you for your note. You are right, we will unify nomenclature and style in a revised document.

**Line 124: what does the flashing?**

**Line 128: which is the range? And the accuracy/precision of the sensor?**

**Line 136: How often did you calibrate? how do you correct?**

Response: Thank you for those questions. The measurement routine was kept brief to avoid disrupting the reading flow. However, we acknowledge that specific terms should be explained, and therefore, we will revise the description of our methods to provide more detailed explanations. We calibrated the probes before and after deployment and took occasional water samples for GC analysis and quality control. In that sense, we will provide additional detailed explanations in the supplement.

**Line 143: How do you transform your 2-m wind to 10-m?**

Response: As mentioned above, we will provide more detailed information on the methods in the revised manuscript. However, the parametrization of U10 was done using the wind.scale() function provided in the LakeMetabolizer package in R. That function is based on a logarithmic wind speed profile described by the equation

$$U10 = Wind \ x \ (\frac{10}{height})^{\frac{1}{7}}$$

**Line 145: Probably nice to know the "maths" behind the function for GPP and R, for example**

Response: Thank you the comment. Explaining all the equations behind the lake metabolism calculations is beyond the scope of our paper. The package LakeMetabolizer, including a GitHub repository with all the functions used, is published in https://doi.org/10.1080/IW-6.4.883. There is also a good review by Peter Staehr about this topic (https://aslopubs.onlinelibrary.wiley.com/doi/10.4319/lom.2010.8.0628).

**Line 150: Out of curiosity (and also for paragraph consistency), do you have measurement of oxygen before the mixing, when temperature was 30°C?**

Response: Yes, we have oxygen data for that period. Oxygen reached up to 170 % saturation during the stratified period. We think that information is out of the scope of this study and it would disrupt the reading flow and distract from the main topic.

**Line 150: The result section needs a bit of re-estructure for clarity. For example, you talk about CO2 during the day in line 159 and again in line 172. I suggest to unify. Same get for other parameters.**

Response: Thank you for your comment. We tried to structure the sections by the content. However, we acknowledge your concerns and will revisit the content structure for a later version of the manuscript.

**Line 155: add in M&M how this equilibrium is calculated. I guess is the CO2 from atmosphere (?). How did you get it?**

Response: Thank you for pointing that out. In fact, the measurement platform was equipped with a CO2 analyzer, which was used to determine EC fluxes too. This is published and cited in our manuscript in

Spank et al. 2020, 2023, and 2024. For the sake of completeness, we will add this to the methods part in the supplement.

**Line 155: consistency: in the main text you use Ta and Tw. Also, do you have the temperature at 5cm?**

Response: Thank you for pointing this out. We will correct the text in general to be consistent throughout. Regarding the temperature in 5 cm depth – we do not have that. We tried to calculate the surface temperature from outgoing radiation measurements. However, this is an indirect measurement, which, further, can be error prone due to waves etc. We therefore decided to use the temperature measured using the thermistor chain, which we found to be the most robust method.

**Figure 1: and at 25 cm?**

Response: This is a very good comment. We did have an oxygen probe installed in 25 cm depth, however, the sensor failed to log data. This was very unfortunate for our study, because it could have helped us resolve the oxygen gradient and the metabolic activities at the surface.

**Figure 1: the bars are a bit misliding, as sometimes there is a lot of variation. I was wondering if it could make more sense to add a box plot, where you will also show the mean and the range**

Response: Thank you for your note. In the process of creating this figure we tried many different designs and found this design to be the best compromise that shows the most important features of our dataset very clearly. Especially the bars helped a lot resolving the diurnal patterns and light dependence of the other parameters.

**Line 170: U 10 m or U 2m? Also mentioned in the figure**

Response: Its U10. We will uniform the nomenclature in the revised version of the manuscript.

Line 175: which is? needs to be stated in the M&M
Response: Thank you for the comment. As mentioned above, this will be part of the revision of the M&Ms part.

**Line 184: I guess the answer is no but do you have measurements of CO2 in the bottom? If not, worth to considered for the next project :)**

Response: Thank you the question and tip. We took monthly water samples for GC analysis from different layers – also from the hypolimnion. Those data show $CO_2$ accumulation in the hypolimnion during stratification – a common observation in lakes (e.g. https://bg.copernicus.org/articles/10/7539/2013/bg-10-7539-2013.pdf). However, in this manuscript we focus on the surface layer.

**Figure 2: Carefull on using deltaCO2 for two different things: air-water (as in Equation 1) or 5-25 cm. Find another abbreviation of metion all the time deltaCO2_(5-25cm)**

Response: Thank you this comment. We did not recognize that before, but will change it for consistency in a revised version of the MS.

Figure 2: This means dots should have a error bars or something that show the variation you are averaging.

Response: Thank you for the comment. We tried different versions of this plot already, including boxplots. Those did not turn out nice and concise. However, adding error bars to the dot is possible and will be considered for the revised version of the manuscript.

Figure 2: See Stolle et al. (2020), figure 13. We presented k but could have also a relationship.

Response: Yes - interesting. It is a common observation that wind speed above lakes is lower in the night. That would mean that the same argumentation (low wind and low wind driven k during the night) applies both to inland water and the sea.

Figure 3: It is always nicer when you can understand the caption and figure without the need to read the paper. Please explain all more precise

Response: Thank you for the comment. We tried keeping the caption short, but we agree with your comment and will make a stand-alone caption in the revised version.

Line 198: careful with the wording, 5 ms-1 is not particularly windy :)

Response: We agree with you – although for a lake we would call 5 m/s rather windy. However, we defined the windy and calm period as mostly windy and mostly calm period, respectively, before. We think this classification is understandable to the reader throughout the manuscript.

Line 205: emission through your study period? We need a time so you can get rid of your time unit.

We are not sure if we understand this comment. We mention the length of our study period here (7 days). We think it is clear that the total emissions given in g/m2 refer to this 7-day period.

Line 207: How relieable are the EC-flux without wind/very low wind speed? You mentioned before they have problems.

Response: The EC data went through a quality control process as referred to and cited in the methods section. Measurements that don't pass the quality control were flagged and withdrawn. Those bad data were excluded and are visible as gaps in the flux plots.

Line 208: The discussion is again very interesting but difficult to read. I suggest to find the main finding of each figure, explaining in the result section and then discussed here. If the journal allow, adding headers sections to the discussion will also help the reader

Response: Thank you for the comment. We agree with you that the discussion can be difficult to read. We already tried to structure it in a way of highlighting the main results and proceed discussing possible causes and interplays. However, we will try to work with sub headers for the discussion section to improve reading flow.

Line 213: in the ocean I measure very low CO2 concentrations but never 0. Is this actually possible/documented? Is like CO2 anoxia?

Response: This is an interesting thought. Due to the carbonate buffering system and CO2 uptake by phytoplankton, very< low CO2 can often be observed in hyper-eutrophic lakes. We do not have pH data

from this study period, but in previous works we observed pH values up to 11. For theoretical reasons $CO_2$ is not zero at this pH, but extremely low. For water-atmosphere diffusion it does not make a big difference whether $CO_2$ is zero or close to zero. For planktonic organisms this is no issue since they rely on bicarbonate. Interestingly, the point of low $CO_2$ concentration is relevant in extremely acidic waters. There, all TIC is present in form of $CO_2$. That means that photosynthetic algae can get carbon limited in such lakes (https://link.springer.com/article/10.1023/A:1005165615804)

**Line 229: I think it is not defined: is like the mixed layer detph in oceanography?**

Response: Thank you for pointing this out. The photo-active layer, also know as euphotic zone, is generally the layer where photosynthesis dominates metabolic processes. It is defined as the part of the mixed layer in which phytoplankton receive enough light to grow. Here, we used the term photo-active layer to decouple it from the term mixed layer, just for the sake of the sentences content. We agree that this was misleading. We will change it to the more common term euphotic zone and describe it.

**Line 247: confusing in the same paragraph, nutrients will help for photosynthesis**

Response: Thank you for that comment. We think that the nutrients that become available after autumn mixing could be one reason for the algae to bloom.

**Line 254: Could we relate that to the slick formation and the effect of CO2 (slow gas-transfer)? If it is what we call a bloom, it should also decrease CO2 via very high photosynthesis.**

Response: We agree with both points. In the discussion, we explain high $CO_2$ demand during the day by the algal bloom. The days were windier than the nights. We therefore think that the formation of a slick mat is favored during the nights. Then this mat would further increase the resistance for gas transfer.

**Figure 4: here you will need SML sampling o microsensors (possible to use in lakes (?)) :). Also nice work from Alexia D. Saint-Macary**

Response: Microsensors are indeed a good method to study gradients at boundaries. We also performed some oxygen microsensor measurements at the water-air interface in the laboratory as well as in a small pond. However, we did not do such measurements in Bautzen. It would be very interesting to try in situ microsensor measurements in lakes in future experiments.

**Figure 4: During day, there is also a phytoplankton CO2 release, just that problably the uptake is bigger.**

Yes – it's a net uptake.

**Also just noticing, sometimes you talk about phytoplankton and sometime algae. In oceanography we use phytoplankton**

Response: Thank you for highlighting this. The correct term would be phytoplankton. We will change algae to phytoplankton in the manuscript to use correct scientific terms.

---

## Author Comment (AC2)

This is a very relevant study as it points to one of the major concerns of the traditional methods to estimate air-water CO2 exchange, the assumption of constant concentration of CO2 in the water. The experiment and data are interesting and well worth publishing. I, however, think that a substantial revision with rewriting and reanalysis is required.

Below are some comments and suggestions, I will not comment much on language, but I think a thorough revision of the writing is necessary as well.

Response: We want to thank you for reading through our manuscript and taking the time to help us improve the work. We are happy to hear that you found it to be a valuable contribution to the field. For a revised manuscript, we will improve the language.

General comments of the introduction:

There exists a range of literature on the role of lakes in the carbon cycle, and this part should be updated base on more and more recent literature. Please look at the papers by Golub et al and Guseva et al for a range of suggested papers.

Response: With our manuscript we focus on CO2 gradients at the water surface, which can affect concentration measurements. This can result in erroneous estimations of fluxes. The focus of our manuscript is not the carbon cycle per se rather the special characteristic of the diffusion at the water surface. We therefore decided to restrict ourselves to 2 references at this point. Of course, there is a lot of literature about the role of lakes in the carbon cycle. The already classical paper by Raymond was the first to review the role of CO2 and in our eyes deserves reference here. We think the Lauerwald reference from 2023 is a good one to present the state of the art.

I think the ambition to look at 5 and 25 cm are good, but the major gradient is closer to the surface, please have a more thorough discussion on this aspect.

Response: We agree that the major gradient is in the surface microlayer. However, with existing instruments it was not possible to monitor $CO_2$ concentrations at the water surface with better vertical resolution. It is indeed one of the messages of our paper that we need a good method to measure dissolve gas concentrations very close to the surface. We will discuss this point in more detail.

Line 35: There are several EC sites worldwide (see Golub and Guseva).

Response: Compared with studies measuring GHG concentrations, there are only "few" EC measurement systems installed on water sites. However, we agree that more and more EC systems are being installed on lakes. We will change "few" to "a restricted number".

Furthermore, we want to emphasize that EC measurements are not suitable for small lakes, or measurements there can be limited due to fetch characteristics and probable interfering of the footprint with the terrestrial and amphibious surrounding. Consequently, CO2 measurements in the water are in many cases the only convenient method to quantify GHG fluxes continuously.

Line 50 to 55: The cooling induced convection and the impact on the gas exchange is very important here (in particularly when seeing the strong diurnal cycle in the result section) and should be further discussed also in the introduction. There exists a range of literature from seas and lakes (Rutgersson and Smedman, 2010, Podgrajsek et al 2015, Eugster et al, 2003, 2023; Heiskanen 2014, Andersson et al 2017)

Response: Thank you for this important comment. Our initial intention was to not go into the details of this topic to direct the readers attention to our main research objectives. However, we agree, especially facing our findings regarding the differences in water temperature and air temperature, that a more detailed discussion of this point can be helpful to understand and discuss our results.

Line 75: The study of Rudeberg is interesting and well discusses how spatial and temporal variations influences the flux. It is, however, limited to chambers. Other studies use EC-fluxes (Rutgersson et al, Dong et al).

Response: Thank you for your comment. In our manuscript we tried to focus on the topic of water side concentrations and its measurements. We acknowledge that a lot of research is available on the spatial variability that is measured as spatial average by EC measurement due to the larger footprint, however, we tried to have a concise story about aquatic concentrations. We therefore introduced and discussed the effects that can be observed when using CO2 probes in the water, rather than focusing on the horizontal heterogeneity of fluxes, which can be measured using EC already. We will include additional sentences on that part in a revised version of the manuscript, while keeping the focus on concentration measurements in the water.

I think the limitations of chambers in relation to EC should be further discussed (see for example Podgrajsek et al 2014).

Response:  As mentioned above, we tried to focus on the limitations of aquatic CO2 concentration measurements. We are aware of the ongoing discussion regarding the comparison of EC and floating chambers and discussed this with many colleagues in the past years. If we start discussing possible issues with floating chambers we also need to discuss possible issues with EC. In our opinion, this is far beyond the focus of this paper and should be discussed in another manuscript. Floating chamber measurements did not play a role in this work – so we not really see the point in discussing floating chambers in detail.

The surrounding areas is considered unimportant, please ale note the possibility of non-local effects (eeg Esters et al).

Response: Thank you for hinting to this topic. Non-local effects were considered unimportant based on comprehensive investigations, personal observations and experience with the water body. In Spank et al (2023) we did a very detailed footprint analysis and showed that the footprint was not affected by littoral areas and the terrestrial surrounding. Furthermore, the analyses of the energy fluxes (i.e., sensible and latent heat flux) provided by the EC measurement system in parallel to the $CO_2$ fluxes clearly showed a unique aquatic characteristic, which additionally proofs the minor importance of non-local effects.  We also did multiple field campaigns to resolve spatial heterogeneity of CO2 concentrations in the surface water of Bautzen Reservoir, but never found hints of pronounced

differences. In particular, we could not find hotspots of CO2 concentrations during the night. The observed concentration gradients are limited to certain nights, for which we have only the data shown in the manuscript.

Section 2.4: Please do not name the routines used. If this is important explain what they do to the data (if this paper is read in 10 years' time, it might be impossible to understand as now written). The name of the routines could be in an appendix, if the authors consider it important information.

Response: We think that it is good practice and important for the reproducibility of the analysis to mention the functions used. We agree that information about the principles behind those functions is interesting. We will reformulate this and move less important stuff to the supplement.

Results:

In Figure 1 you show a really nice diurnal cycle, with significant gradients during night-time, this is explained by the phytoplankton activity, but you really should consider the effect of physical processes with a strong waterside convection during night-time. This is seen during low winds, when the convection is found to dominate.

Response: Thank you for your comment. In preparation of the manuscript we tried to investigate processes that could explain the observed CO2 gradient, while having condition that favor convection. However, both the atmosphere and the water were in unstable condition, which is derived the difference between water and air temperature, the course of the water temperature profile in surface water and stability measures provided by the EC measurements, but we did not find a robust explanation for the differences in CO2. We agree that our discussion of night-time convection is a bit short. In a revision we will discuss this aspect in more depth, including references recommended by the reviewer.

Andersson, A., E. Falck, A. Sjöblom, N. Kljun, E. Sahlée, A. M. Omar, and A. Rutgersson (2017), Air-sea gas transfer in high Arctic fjords, Geophys. Res. Lett., 44, doi:10.1002/2016GL072373.

Dong et al 2021, https://doi.org/10.5194/acp-21-8089-2021

Esters L, Rutgersson A, Nilsson E and Sahlee E 2020 Non-local impacts on eddy-covariance air-lake $CO_2$ fluxes *Bound.-Layer Meteorol.* **178** 283–300

Eugster W *et al* 2003 CO$_2$ exchange between air and water in an arctic Alaskan and midlatitude Swiss lake: importance of convective mixing *J. Geophys. Res.* **108** 4362

Eugster W, DelSontro T, Shaver G R and Kling G W 2020 Interannual, summer, and diel variability of CH$_4$ and CO$_2$ effluxes from Toolik Lake, Alaska, during the ice-free periods 2010–2015 Environ. Sci.: Process. Impacts 22 2181–98

Golub et al., 2023 Diel, seasonal, and inter-annual variation in carbon dioxide effluxes from lakes and reservoirs, Environ. Res. Lett. 18 034046, https://doi.org/10.1088/1748-9326/acb834

Heiskanen J J, Mammarella I, Haapanala S, Pumpanen J, Vesala T, Macintyre S and Ojala A 2014 Effects of cooling and internal wave motions on gas transfer coefficients in a boreal lake Tellus B 66 22827

Guseva, S., etal , 2023. Bulk Transfer Coefficients Estimated from Eddy-Covariance Measurements over Lakes and Reservoirs. J Geophys Res.-Atmospheres, 128, e2022JD037219, doi:10.1029/2022JD037219. https://agupubs.onlinelibrary.wiley.com/doi/10.1029/2022JD037219

Podgrajsek E, Sahlée E and Rutgersson A 2015 Diel cycle of lake-air CO$_2$ flux from a shallow lake and the impact of waterside convection on the transfer velocity *J. Geophys. Res.* **120** 29–38

Rutgersson A. and Smedman, A. Enhancement of CO2 transfer velocity due to water-side convection, *J. Marine Syst*., 80, 125-134,. 2010

Rutgersson, A., M. Norman, B. Schneider, H. Pettersson, E. , Sahlée. The annual cycle of carbon-dioxide and parameters influencing the air-sea carbon exchange in the Baltic Proper. *J. Mar. Syst.,* 74, 381-394. 2008